# Conversion of anilines to chiral benzylic amines via formal one-carbon insertion into aromatic C–N bonds

Lei Li[1], Min Yang[2], Qiuqin He[1] & Renhua Fan [1✉]

Insertion of atoms into aromatic carbon-nitrogen bonds is an appealing method for the synthesis of nitrogen-containing molecules and it has the advantage of the availability and abundance of anilines. However, the direct cleavage of aromatic carbon-nitrogen bonds is challenging due to the particularly inert and stable nature of these bonds. Here we report a formal, enantioselective one-carbon insertion into an aromatic carbon-nitrogen bond via an aromaticity dissembly-reconstruction process to directly convert anilines to chiral α-branched benzylic amines. The process involves oxidative dearomatization of para-substituted anilines, chiral sulfur ylide-mediated asymmetric aziridination, and subsequent rearrangement. Chiral sulfur ylides serve as one-carbon insertion units.

[1] Department of Chemistry, Fudan University, 200433 Shanghai, China. [2] Department of Forensic Science, Oil-tea in Medical Health Care and Functional Product Development Engineering Research Center in Jiangxi, Gannan Medical University, Ganzhou 341000, China. ✉email: rhfan@fudan.edu.cn

Insertion of atoms into chemical bonds is an attractive transformation of organic molecules because it leads to the simultaneous formation of two new chemical bonds. In recent decades, significant progress has been made in transition-metal-catalyzed insertion of atoms into unreactive chemical bonds such as carbon–carbon[1–3], carbon–cyanide[4,5], and aliphatic carbon–nitrogen bonds[6–10]. The insertion reaction normally involves transition-metal-catalyzed cleavage of the bond in question and subsequent insertion of unsaturated units, such as alkenes[11–15], alkynes[16–18], 1,3-dienes[19,20], or carbenoids[21] (Fig. 1a). These elegant reactions have emerged as an attractive approach to rapid building of complex structures from readily available starting materials. In this context, insertion of atoms into aromatic carbon–nitrogen bonds is an appealing method for the synthesis of nitrogen-containing molecules and has the advantage of the availability and the abundance of anilines. However, the direct cleavage of aromatic carbon–nitrogen bonds is challenging due to the particularly inert and stable nature of these bonds[22,23]. Although aromatic carbon–nitrogen bonds can be activated by converting anilines to more reactive intermediates such as aryldiazonium salts[24–27], arylammonium salts[28–30], amides[31,32], or triazenes[33], the nitrogen atom is usually discarded in byproducts (Fig. 1b).

Chiral α-branched benzylic amines are important structural motifs found in a wide range of natural products and biologically active compounds[34–37]. Driven by the value of active pharmaceutical ingredients, the asymmetric arylation of aldimines by arylmetallic reagents, including lithium[38,39], zinc[40], titanium[41], tin[42], and boron reagents[43,44], has been established as an efficient method for the synthesis of enantiopure benzylic amines. Enantioselective one-carbon insertion into the aromatic carbon–nitrogen bonds is an appealing route with which to establish nitrogen-substituted benzylic stereocenters, and this reaction could satisfy an unmet need in reaction design (Fig. 1c).

Here, we report a formal, enantioselective aromatic carbon–nitrogen bond one-carbon insertion reaction that converts an aniline to a highly functionalized chiral α-branched benzylic amine via an aromaticity dissembly-reconstruction process. The process involves oxidative dearomatization of para-substituted anilines, chiral sulfur ylide-mediated asymmetric aziridination, and subsequent rearrangement. Chiral sulfur ylides serve as the one-carbon insertion unit.

## Results

**Initial tests**. In connection with our recent works on the dearomatization conversion of anilines, oxidative dearomatization can transform the aromatic carbon–nitrogen bond in anilines to a carbon–nitrogen double bond by destroying the aromatic system[45–49]. As shown in Fig. 2, chiral sulfur ylides might serve as a one-carbon unit to be introduced through asymmetric aziridination[50–55]. Subsequent rearrangement might be promoted by a Brønsted or Lewis acid to redevelop the aromaticity and complete the formal enantioselective one-carbon insertion. This builds the nitrogen-substituted benzylic stereocenter and is accompanied by migration of the para-substituent to the meta position and concomitant para-substitution by a nucleophilic reagent. To implement this strategy, the rapid oxidative dearomatization of p-toluidine 1 was tested by examining various oxidants including mCPBA, AcOOH, t-BuOOH, $H_2O_2$, PhI(OAc)$_2$, and PhIO. PhIO together with methanol as the solvent proved to be the best oxidation conditions for the dearomatization. After removing the methanol in vacuo, the crude dearomatized product was mixed with a solution of achiral sulfonium salt S1 in acetonitrile in the presence of 1.2 equivalents of NaH, followed by the treatment with 2 equivalents of CF$_3$COOH (Fig. 3). To our delight, the aziridination and subsequent rearrangement proceeded smoothly and delivered benzylic amine 2 in good yield (71%).

**Optimization of reaction conditions**. Encouraged by this result, a chiral sulfonium salt S2 derived from the Metzner sulfide[56,57] was employed, and this reaction gave rise to compound 2 with 51% yield and 17% ee. Various chiral sulfonium salts were examined[58–64], and S8, derived from isothiocineole[65–68], was observed to be the best chiral ylide precursor, leading to a 64% yield and 95% ee of 2. Changing the anion of sulfonium salts has an influence on the yield of 2 but leads to no change in the enantioselectivity. Reaction with sulfonium trifluoromethanesulfonate S10 provided 2 in 78% yield and 94% ee. Changing the nature of protecting group on the aniline nitrogen atom has a major effect on the transformation. The reaction works well with various sulfamide groups but not with acetamide or benzamide. A set of reaction variables including bases, Lewis and Brønsted acids, solvents, temperatures, and the ratio of reagents were investigated to establish the optimum reaction conditions. NaH proved to be the best base. A variety of Brønsted or Lewis acids shown different activities to promote the rearrangement. When 20 mol% Cu(OTf)$_2$ was used instead of 2 equivalents CF$_3$COOH, the yield of 2 was improved to 79% yield and 96% ee (for details, see Supplementary Table 1 in the Supplementary Information).

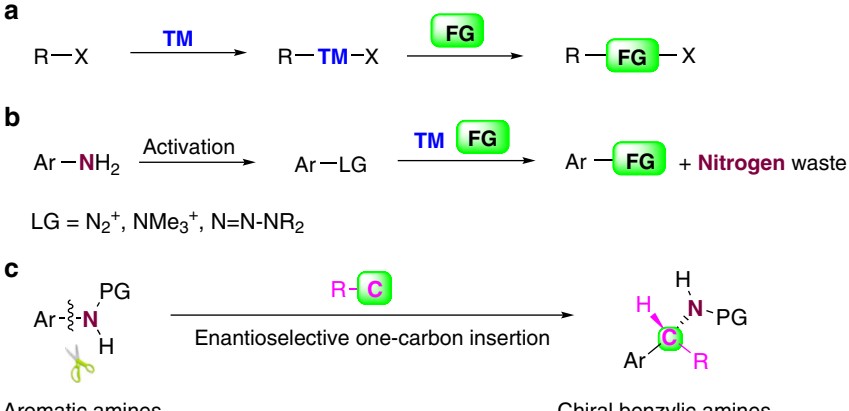

**Fig. 1 Atoms insertion and transformation of aromatic N–C bonds. a** Transition-metal-catalyzed atoms insertion. **b** Conventional transformation of aromatic nitrogen-carbon bonds. **c** One-carbon insertion into aromatic nitrogen-carbon bonds. TM transition metal, FG functional group, LG leaving group, PG protecting group.

**Fig. 2 Formal one-carbon insertion into aromatic carbon–nitrogen bonds.** The transformation might proceed via a dearomatization of *para*-substituted anilines, an chiral sulfur ylides-mediated asymmetric aziridination, followed by a rearrangement to recover the aromaticity and complete the formal enantioselective one-carbon insertion accompanied by a migration of the *para*-substituent to the meta position and *para*-substitution by a nucleophilic reagent. PG protecting group, Nuc nucleophile.

**Fig. 3 Evaluation of sulfur ylides.** nd not determined.

**Scope of anilines**. The scope of this transformation was investigated by systematically varying the anilines and the chiral sulfonium salts. As shown in Fig. 4, reactions of a range of anilines with the chiral benzyl sulfonium salt **S10** proceeded smoothly. In addition to the methyl group, ethyl, n-butyl, isopropyl, cyclohexyl, or phenyl groups can be the *para*-substituent in the anilines and migrate to the meta position in the product. An electronic effect was observed for anilines with different meta-substituents. For example, anilines with a meta-phenyl group gave **9** in higher yield than the reaction with a meta-methyl group, and aniline bearing a meta-4-fluorophenyl group gave **12** in higher yield than that bearing a meta-4-methoxyphenyl group. The *ortho*-substituent of anilines have an effect on the reaction. When 2-fluoro-4-methylaniline was employed, the 4-methyl group migrated to the C-3 position, leading to the formation of **17** in 61% yield and 99% ee. When 2-bromo-4-methylanilines were used, the reaction gave rise to a mixture of the corresponding C-3 or C-5 migration products **18**. Reaction of 4-methyl-2-(phenylethynyl)aniline also provided a mixture of the C-3 or C-5 migration products **19**. The formation of the mixture of migration products might be caused by the combined influence of the electron-withdrawing and steric effects of the *ortho*-substituent. The electron-withdrawing effect makes the C-3 position more positively charged compared to the C-5 position, and the steric effect makes the C-3 position more hindered than the C-5 position.

Due to the strong electron-withdrawing property and the small size of the fluorine atom, the C-3 position is the preferred site for the migration of the 4-methyl group. Therefore the reaction of 2-fluoro-4-methylaniline only produced the C-3 migration product **17**. When the *ortho*-substituent is a bromine atom or a phenylethynyl group, the relatively weaker electron-withdrawing property and the bigger size led to the formation of mixture of the C-3 or C-5 migration products. Different functional groups can be introduced into the *para*-position by varying the solvent used in the dearomatization step. For example, the use of ethanol, isopropanol or trifluoroethanol instead of methanol led to the formation of the 4-ethoxy, the 4-isopropoxy, or the 4-trifluoroethoxy-substituted products **20–22**.

**Substrate scope of sulfonium salts**. A wide range of substituents, such as alkyl, methoxy, halogens, trifluoromethyl, phenyl, ester, or boron functional groups on the aryl group of chiral benzyl sulfonium salts, is tolerated under the reaction conditions (Fig. 5). The reaction proceeds smoothly and independently of the different electronic demands on the aryl substituents of chiral benzyl sulfonium salts. For example, the reaction involving a 4-methylphenyl or 4-(trifluoromethyl)phenyl gave rise to compound **25** in 79% yield with 95% ee and **35** in 76% yield with 92% ee. Changing the

*a*3 equivalents of CF$_3$COOH was used instead of Cu(OTf)$_2$. *b*20 mol% of Sc(OTf)$_3$ was used instead of Cu(OTf)$_2$.

**Fig. 4 Scope of anilines.** Ts 4-toluene-sulfonyl.

substituent position in benzyl sulfonium salts affected the reaction yield and enantioselectivity. For example, the 2-methoxy substituted benzyl sulfonium salt gave compound **26** in 91% yield with 96% ee, and the 4-methoxy substituted salt gave compound **28** in 58% yield with 92% ee. The reaction of a sulfonium salt bearing two meta-tert-

butyl groups provided **41** in 77% yield with 95% ee. When furan-3-ylmethyl sulfonium salts were employed, the reaction provided α-furan substituted benzylic amine **43** in 59 yield with 80% ee. When α-unsubstituted allyl and propargyl sulfonium salts were employed, the corresponding α-branched benzylic amines **45** and **46** were

**Fig. 5 Scope of chiral sulfonium salts.** Ts 4-toluene-sulfonyl.

[a]20 mol% of Sc(OTf)$_3$ was used instead of Cu(OTf)$_2$. [b]3 equivalents of CF$_3$COOH was used instead of Cu(OTf)$_2$.

formed in moderate yields but with lower enantioselectivity. As the steric hindrance of the allyl sulfonium salts is increased by incorporation of an α-methyl substitutent, the stereochemical control of the reaction increases markedly leading to the formation of **46** in 96% ee, but the yield decreased. When the steric hindrance of anilines is increased by incorporation of an *ortho*-methyl substitutent, product **47** was not formed because the corresponding aziridination reaction did not occur. The absolute configuration of **34** was confirmed by X-ray crystallography (see Supplementary Fig. 1 in the Supplementary Information).

**Mechanistic studies**. To gain more insight into the transformation, the corresponding dearomatized intermediate **48** and the azidination intermediate **49** were isolated. Both of them can be converted into product **2** under the standard conditions (Fig. 6a). When unsubstituted benzenamine was employed as the substrate, 2.1 equivalents of PhIO were required to facilitate the oxidative dearomatization to generate a quinone imine ketal **51** as the intermediate. However, the reaction of **51** under the standard conditions gave rise to N-(4-methoxyphenyl)-4-methylbenzenesulfonamide **52** instead of the insertion product (Fig. 6b). When two aziridination intermediates **53** and **54** were mixed and treated with Cu(OTf)$_2$, the reaction gave rise to compounds **55** and **56**, and the formation of **57** and **58** was not observed (Fig. 6c). This result indicated the migration of the *para*-alkyl group proceeds via an intramolecular manner.

**Fig. 6 The mechanistic study. a** Reactions of the isolated intermediates. **b** Reaction of unsubstituted benzenamine under the standard conditions. **c** Reaction of a mixture of two aziridination intermediates.

**Fig. 7 Plausible reaction pathway.** Path a is the preferred way for the formation of the one-carbon insertion product accompanied by a migration of the *para*-substituent to the meta position and concomitant *para*-substitution by a nucleophilic reagent.

A plausible pathway for this transformation was depicted in Fig. 7. PhIO mediates the oxidative dearomatization of *para*-substituted anilines in methanol to generate cyclohexadienimines. The nucleophilic addition of chiral sulfur ylides to cyclohexadienimines and subsequent cyclization lead to the generation of the spiro intermediates. Rearomatizing to release the tension of the spiro structure is a great driving force for the rearrangement. With the aid of a Brønsted or Lewis acid, the migration of the alkyl or the aryl group forms intermediate **I** (path a), while the migration of the methoxy group forms intermediate **II** (path b). Because the positive charge in intermediate **I** can be stabilized by the oxygen atom (intermediate **III**), rearrangement via path a is preferred. Final aromatization delivers the one-carbon insertion products. When the substituent at the *para*-position of amino group was a methoxy group, dearomatization occurred to give an acetal intermediate, but aziridination and C–C bond cleavage were not observed under standard conditions.

**Synthetic applications.** The reaction magnifying 50 times can occur normally which can get product in 77% yield and 94% ee (Fig. 8a). Transformations of the one-carbon insertion products have been explored. For example, under reductive conditions, the N-tosyl protecting group of compound **2** can be removed, leading to the formation of N–H free chiral benzylic amine **48** (Fig. 8b). Compound **23** bearing an *ortho*-methyl group undergoes a radical sp³ C–H amination reaction to form a 1-substituted isoindoline **49** (Fig. 8c). Compound **33** with an *ortho*-iodo group is readily converted to the 1,3-disubstituted isoindoline **50** via a Pd-catalyzed cascade coupling/cyclization reaction (Fig. 8d). In these transformations, the enantiopurity of the substrates is preserved in the products.

## Discussion
In summary, we report a formal enantioselective aromatic carbon–nitrogen bond one-carbon insertion reaction via an aromaticity dissembly-reconstruction process to directly convert

**Fig. 8 Reaction in a gram scale and transformations of the one-carbon insertion products. a** Reaction in a gram scale. **b** Removal of the N-tosyl protecting group of compound **2**. **c** Radical amination reaction of compound **23** to form 1-substituted isoindoline. **d** Pd-catalyzed cascade coupling/cyclization reaction of compound **33** to form 1,3-disubstituted isoindoline.

anilines to chiral α-branched benzylic amines. The process involves three steps: oxidative dearomatization to activate the aromatic carbon–nitrogen bond in *para*-substituted anilines by breaking the aromatic system, chiral sulfur ylide-mediated asymmetric aziridination to introduce the one-carbon unit, and subsequent rearrangement to recover the aromaticity and complete the formal enantioselective one-carbon insertion accompanied by a migration of the *para*-substituent to the meta position and concomitant *para*-substitution by a nucleophilic reagent. Development of extensions of this group insertion strategy to other aromatic systems is currently in progress in this laboratory.

## Methods

**Representative procedure**. PhIO (0.22 mmol) was added to a solution of *N*-Ts *p*-toluidine 1 (0.2 mmol) in MeOH (2.0 mL) at 25 °C. After 5 min, the reaction mixture was concentrated in vacuo. The resulting mixture was mixed with the sulfonium salt (0.24 mmol) and sodium hydride (0.24 mmol) in MeCN (2 mL). The reaction was stirred at rt for 3 h, then Cu(OTf)$_2$ (0.02 mmol) was added. After the intermediate was completely consumed (monitored by TLC analysis), the reaction was quenched with saturated NaHCO$_3$ (25 mL), and extracted with EtOAc (25 mL × 3). The organic layer was dried over Na$_2$SO$_4$, and concentrated in vacuo. The residue was purified by flash column chromatography on silica gel (eluent: petroleum ether/EtOAc) to furnish the desired compound **2**. White solid; mp: 105–106 °C; ${}^1$H NMR (400 MHz, CDCl$_3$): δ 7.54 (d, *J* = 8.2 Hz, 2H), 7.22–7.17 (m, 3H), 7.15–7.10 (m, 4H), 6.84 (dd, *J* = 8.4, 1.9 Hz, 1H), 6.80–6.72 (m, *J* = 2.0 Hz, 1H), 6.62 (d, *J* = 8.4 Hz, 1H), 5.48 (d, *J* = 7.1 Hz, 1H), 5.23 (d, *J* = 7.1 Hz, 1H), 3.75 (s, 3H), 2.37 (s, 3H), 2.05 (s, 3H); ${}^{13}$C NMR (101 MHz, CDCl$_3$) δ 157.1, 143.0, 140.8, 137.4, 132.1, 129.7, 129.2, 128.4, 127.3, 127.2, 126.7, 125.8, 109.6, 60.9, 55.3, 21.4, 16.1; HRMS (m/z): [M + Na]$^+$ calcd. for C$_{22}$H$_{23}$NO$_3$S, 404.1291; found, 404.1291. The ee value was determined by HPLC

analysis: Chiralcel OD-H Column, hexane/2-propanol = 90/10, 25 °C, 1.0 mL/min, 220 nm, retention time: 15.8 min (minor) and 21.3 min (major).

## Data availability

All data that support the findings of this study are available within this article and its Supplementary Information (including experimental procedures and compound characterization data). The X-ray crystallographic coordinates for structure 34 reported in this study has been deposited at the Cambridge Crystallographic Data Centre (CCDC), under deposition numbers 1966589. These data can be obtained free of charge from The Cambridge Crystallographic Data Centre via www.ccdc.cam.ac.uk/data_request/cif. Data are also available from the corresponding author upon reasonable request.

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

## Acknowledgements

We are grateful to the National Natural Science Foundation of China (21572033 and 21971043) and the Science and Technology Commission of Shanghai Municipality (18XD1400800 and 19ZR1403400) for support of this research.

## Author contributions

R.F. directed the research and developed the concept of the reaction with L.L. and M.Y., who also performed the experiments and prepared the Supplementary Methods. Q.H. checked the experimental data. R.F. wrote the manuscript with contributions from the other authors.

## Competing interests

The authors declare no competing interests.
