## [Peer Review File · Nature Communications]

Reviewers' Comments:

Reviewer #1:

Remarks to the Author:

The paper "Enantioselective One-Carbon Insertion into Aromatic C-N bonds via Dearomatization: Conversion of Anilines to Chiral Benzylic Amines" attempts to describe a truly novel method to prepare chiral benzylic amines. The benzylic amines are generated after aziridination of an oxidized aniline with a chiral sulfonium ylide followed by an acid catalyzed rearrangement to give good yields and high enantioselectivities. Bizarrely, the manuscript does not discuss the oxidation conditions nor the acid-catalyzed rearrangement conditions. The transformation itself is novel, but given the lack of a detailed mechanistic investigation, does not offer much to the field of organic synthesis. The preparation of chiral benzylic amines is already well-established despite this manuscript's failure to make any mention of these methods. Many of these methods are catalytic in chiral material in stark contrast to the current work which employs 1.2 equivalents of a chiral sulfur ylide. The ylide is limited in scope, too, only providing products which are either bis-benzylic, benzylic-allylic, or benzylic-propargylic with only four total examples of the latter two categories. The aniline starting material is also limited, requiring a para-alkyl substituent (for reasons which were not discussed) and incorporating a para-methoxy substituent during the process (shifting the para-alkyl substituent to a para-meta substituent). The requirement of a functional handle at a relatively distal location hampers the applicability of this method even before considering the uncontrollable alkyl-shift and para-heteroatom incorporation from a molecule of solvent during the fully undiscussed oxidation step.

Despite having a relatively detailed and well-put-together SI, the manuscript itself falls well short of even describing the work that was done. The transformation is genuinely interesting in and of itself, but the authors choice to focus on environmental friendliness and straightforwardness of the method certainly does not help them make any sort of case in favor of the method. The lack of context, mechanistic study, and discussion of two steps of the three-step protocol serve to hinder the reader's understanding of the method, which as discussed above, is limited in application. Additionally, several extraordinary claims are made without any citations, and appear to be meant to be taken as self-evidently true. Far from being self-evident, these claims are contradicted by the very work that is described here (see below for specific examples). In addition to describing a highly specific method with limited general applications, the authors have prepared a manuscript which is only slightly intelligible in its claims and which omits many key features in its short discussion. It is for the limitations of the method that this reviewer recommends submission to another journal. Despite the novelty of the transformation, the lack of mechanistic study in this body of work along with the existence alternative methods for preparing the same substrates limit its usefulness to the synthetic community. Regardless of where this work is eventually submitted, major revisions to the manuscript are needed for it to be acceptable as a scientific document.

Manuscript:

Abstract: Line 14-16, "Insertion of functional groups into chemical bonds" is vague in meaning and "ecofriendly transformation of organic molecules.. without significant generation of byproducts" does not appear to be necessarily true, even though it is claimed to be without citation. It should be noted that the method described employs 1 eq of PhIO and 1.2 eq of a chiral sulfonium salt which fatally undercuts the claim that this method is especially and self-evidently environmentally benign.

Line 17: "insertion of atoms into aromatic carbon-nitrogen bonds is one of the most direct methods for the synthesis of nitrogen-containing aromatic molecules" This is an extraordinarily broad claim for such a specific method. And "most direct" is not defined in any way.

Line 25: Omit the word method.

Lines 33-35 do not cite any source for the veracity of this claim. The sentence itself implies the claim is self-evident, but it is not even true for this very method let alone a general principle.

Line 36: Sometimes the authors say "insertion of functional groups" or just "groups" and sometimes it is "atoms." This should be changed for the sake of consistency.

Lines 42-45: Once again, the claim is asserted as a brute-force and/or self-evident fact. The

authors appear to be claiming that their method is the "most direct" way of making "nitrogen-containing aromatic molecules" despite the fact that both their starting materials and products are "nitrogen-containing aromatic molecules." The claim does not even meet the standard of being consistent with the rest of the paper even if it did have external references to validate it, which it does not.

Line 45 needs the word "the" prior to the word "aromatic"

Lines 52-53: the phrase "after further transformations" confuses the intended meaning of the sentence.

Figure 1: It may be informative to readers to include methods for making chiral benzylic amines so those who may wish to use this method can easily grasp the advantages. Regardless, alternative methods to chiral benzylic amines must be listed in the text or it will be assumed by the readers that the authors are intentionally discouraging a direct comparison to known methods.

Line 66: This sentence means nearly nothing and could be safely removed.

Line 67: The language here is imprecise and should be changed. The carbon-nitrogen bond is not activated by dearomatization. A new carbon-nitrogen (π) bond is created by the dearomatization, which can engage the sulfur ylide. This is a misleading shorthand disguised as a formality.

Line 70: This is the first time it is called a "formal" insertion, but likely it would be more honest to call it a "formal" insertion from the beginning. The fact that it is a formal insertion which relies on oxidative dearomatization to proceed makes Figure 1a seem even more out of place than it already did.

Line 71: Should read "This builds..."

Line 81: "In turn" adds confusion.

Line 82: An effect of reactivity that is caused by "electronic and steric factors" is so broad as to include every possible factor. The sentence therefore reads in an identical way as the following: "The ketimino group might be less reactive because of something." This need to be corrected.

Line 82: "Our initial of the viability" does not make grammatical sense, maybe from a word mistakenly being omitted.

Line 84: The authors, when describing their initial viability studies, erroneously fail to mention the oxidation step of p-toluidine saying, "reaction of p-toluidine 1 with an ylide." This simply is not what was done in the study shown, is inconsistent with the proposed mechanism, and must be corrected.

Line 95: The authors list variables tested and, once again, fail to mention anything about the oxidant. It's necessity in the reaction, for example, must be included. This same logic also applies as completely to the acid-catalyzed rearrangement, which is also erroneously missing from all discussion.

Line 106: The word "they" before "migrate" is not necessary and should be omitted.

Line 119: "but" should be "and" for the sentence to make the most sense.

Line 121: Dearomatization step is finally mentioned, albeit in passing.

Line 136: Please change to "The reaction of a sulfonium.."

Line 139: Strange symbol typo in this reviewer's copy.

Line 142: same problem as line 139. Looks like a problem with the alpha symbol.

Line 153: "the synthesis of nitrogen-containing aromatic compounds"- both starting materials and products for all of these examples meet this definition. Following this logic, the most direct way to prepare a nitrogen containing aromatic compound is to simply already have one.

Line 176: Along with a discussion of the oxidants screened, mention of the requirement of a substituted aniline as a starting material is omitted but should not be.

Line 176: There is similarly no discussion whatsoever of the role of copper in the reaction. There is no discussion of the necessity of copper in the reaction. The word copper is not even included anywhere in the text. A discussion of the acids for the promotion of the rearrangement is not mentioned anywhere in the text.

Line 176: The removal of solvent after the oxidation is also ignored in the main text. This is a multi-step method, and the authors are not served by ignoring the actual protocol.

SI:

Formatting issues in the TOC need to be addressed

The following errors occur relatively routinely, and the entirety of the SI should be checked again:

1. The counterion in the MS calc. data is not included in the calculated formula in any of the substrates but should be. For instance $[M+Na]^+$ calcd. for $C_{27}H_{31}NO_3SNa$ (sodium atom included).
2. Multiple substrates have too few carbon signals. HSQC and HMBC should let you know if there are two overlapping, and the fact of a single signal for two different carbon peaks should be noted.

Reviewer #2:

Remarks to the Author:

Fan and coworkers demonstrated an interesting formal aromatic carbon-nitrogen insertion process for asymmetric synthesis of benzyl amines. The key to this process relied on the novel oxidative nucleophilic dearomatization/asymmetrical sulfur-ylide initiated aziridination/migratory aromatization sequence. The scope of this reaction is reasonable broad, providing chiral diphenylmethanamine derivatives in high yields and ees. Given this novel reaction design and synthetic useful products, the referee would recommend the acceptance of this manuscript for publication in Nat Commun.

1)When ortho-substituted anilines were used, 3-substituted products were dominated in comparison with 5-substituted products. Could the authors explain the reason for the unusual selectivity? Why did the migration happen at a bulkier site? In addition, there was lack of explanation as fluorinated substrate 17 only led to single regiomer.

2)In this manuscript, only semistable sulfur ylides have been tested. The stable ylides bearing an ester or ketone group, are very interesting substrates for this transformation as they will provide chiral phenylglycines.

Reviewer #3:

Remarks to the Author:

I was very interested to read this novel contribution from Fan and co-workers. The manuscript describes the transformation of readily available anilines into benzylic amines with insertion of a carbon between the nitrogen and aromatic ring of aniline. In addition there is an addition of a nucleophile to the para-position of the aniline with migration of the p-substituent to the meta-position, thus adding further complexity to the product. Using a readily available chiral sulfide the process has been rendered highly enantioselective. I think the paper will be of significant interest to the chemistry community and should be accepted following minor revisions (see below).

The dedication to Prof Dai is most fitting.

The reactions have been carried out at a relatively small scale. It is important that one example is reported to make 1 g of product to demonstrate the robustness of what appears to be a very useful and interesting reaction.

With respect to the proposed reaction pathway, have the authors evidence for the formation of the proposed aziridine intermediate? E.g., can it be isolated by omitting the final step in the sequence?

Line 90: the authors mention the choice of counterion affects the yield of the reaction. Since the reaction proceeds via the ylide (with no counterion) I wonder whether the yield is impacted by the solubility of starting salt. Perhaps the authors could comment.

Fig. 3 The diagram shows a range of ylides derived from literature known chiral sulfides. Literature references should be provided for each of these sulfides (not currently the case) and an additional reference should be supplied for the best sulfide (Synthesis 2018, 3337 by Aggarwal).

The way of drawing Aggarwal's sulfide might be misleading with respect to the dimethyl

substitution at the carbon alpha to sulfur. It would make it less obvious that approach to one face of the ylide is blocked. Consider changing.

Line 105: in relation to substituents tolerated at the para-position of the aniline, are alkene and alkyne groups tolerated?

Fig.5 The caption refers only to 'benzyl' sulfonium salts but actually the scope goes beyond benzyl (which is excellent). I think the caption should be re-worded to capture this.

Line 146,147: it would be worth noting whether the absolute configuration obtained is in line with expectations based on the literature model for enantioselectivity in S-Ylide aziridinations.

Presumably all other configurations are assigned by analogy – a statement to this effect could be included in the supporting information.

There are no references in the Supporting Information. There should be some statements about the sources of starting materials and where these are not commercially available a literature reference should be provided for their synthesis e.g., the various sulfonium salts used. Were all of the anilines commercially available? Are all of the products novel? If not, then references should be provided and a statement that literature data matched should be included.

Other than that the supporting information appears to be excellent with appropriate characterization and copies of spectra and chromatograms included.

Here are the details on the point-by-point response to the referees' comments:

Answers for Comments of reviewer 1:

1. Comment:

Bizarrely, the manuscript does not discuss the oxidation conditions nor the acid-catalyzed rearrangement conditions. The transformation itself is novel, but given the lack of a detailed mechanistic investigation, does not offer much to the field of organic synthesis.

Answer:

The discussion on the details of the oxidation conditions and the acid-catalyzed rearrangement conditions have been added:

Page 4: "To implement this strategy, the rapid oxidative dearomatization of *p*-toluidine **1** was tested by examining various oxidants including *m*CPBA, AcOOH, *t*-BuOOH, H₂O₂, PhI(OAc)₂ and PhIO. PhIO and methanol as the solvent proved to be the best oxidation conditions for the dearomatization. After removing the methanol in vacuo, the crude dearomatized product was mixed with a solution of achiral sulfonium salt **S1** in acetonitrile in the presence of 1.2 equivalents of NaH, followed by the treatment with 2 equivalents of CF₃COOH (Fig. 3). To our delight, the aziridination and subsequent rearrangement proceeded smoothly and delivered benzylic amine **2** in good yield (71%)." **was added.**

Page 3: "Subsequent rearrangement can redevelop the aromaticity and complete the formal enantioselective one-carbon insertion." **was replaced by** "Subsequent rearrangement might be promoted by a Brønsted or Lewis acid to redevelop the aromaticity and complete the formal enantioselective one-carbon insertion."

Page 4: "In this way, the yield of **2** was improved to 79% yield and 96% ee" **was replaced by** "NaH proved to be the best base. A variety of Brønsted or Lewis acids shown different activities to promote the rearrangement. When 20 mol% Cu(OTf)₂ was used instead of 2 equivalents CF₃COOH, the yield of **2** was improved to 79% yield and 96% ee".

The mechanistic study was added. Please see the Answer to the Comment 6.

2. Comment:

The preparation of chiral benzylic amines is already well-established despite this manuscript's failure to make any mention of these methods. Many of these methods are catalytic in chiral material in stark contrast to the current work which employs 1.2 equivalents of a chiral sulfur ylide. The ylide is limited in scope, too, only providing products which are either bis-benzylic, benzylic-allylic, or benzylic-propargylic with only four total examples of the latter two categories.

Answer:

The introduction of the preparation of chiral benzylic amines and the corresponding references were added:

Page 2: "For example, enantioselective one-carbon insertion into aromatic carbon-nitrogen bond of anilines is an appealing route for the synthesis of enantiopure α -branched benzylic amines, an important structural motif in natural products and active pharmaceutical ingredients⁴⁰⁻⁵³" **was deleted.**

Page 2: "The enantioselective insertion of a one-carbon unit into aryl carbon-nitrogen bonds remains an elusive challenge (Fig. 1c)." **was deleted.**

Page 3: "Chiral α -branched benzylic amines are important structural motifs found in a wide range of natural products and biologically active compounds.¹³ Driven by the value of active pharmaceutical ingredients, the asymmetric arylation of aldimines by arylmetallic reagents, including lithium,¹⁴ zinc,¹⁵ titanium,¹⁶ tin,¹⁷ and boron reagents¹⁸ has been established as an efficient method for the synthesis of enantiopure benzylic amines. Enantioselective one-carbon insertion into aromatic carbon-nitrogen bonds is an appealing route with which to establish nitrogen-substituted benzylic stereocenters, and this reaction could satisfy an unmet need in reaction design (Fig. 1c)." **was added.**

The corresponding references **was added** as references 62-83.

3. Comment:

The aniline starting material is also limited, requiring a para-alkyl substituent (for reasons which were not discussed) and incorporating a para-methoxy substituent during the process (shifting the para-alkyl substituent to a para-meta substituent). The requirement of a functional handle at a relatively distal location hampers the applicability of this method even before considering the uncontrollable alkyl-shift and para-heteroatom

incorporation from a molecule of solvent during the fully undiscussed oxidation step.

Answer:

Reaction of unsubstituted benzenamine was added:

Page 13: Unsubstituted benzenamine was employed, but unfortunately, byproduct without atom insertion was gotten (Fig. 6b).

4. Comment:

Despite having a relatively detailed and well-put-together SI, the manuscript itself falls well short of even describing the work that was done. The transformation is genuinely interesting in and of itself, but the authors choice to focus on environmental friendliness and straightforwardness of the method certainly does not help them make any sort of case in favor of the method. The lack of context, mechanistic study, and discussion of two steps of the three-step protocol serve to hinder the reader's understanding of the method, which as discussed above, is limited in application.

Answer:

The discussion of the step of oxidative dearomatization and the step of rearrangement was added. Please see the Answer to the Comment 1.

The mechanistic study was added. Please see the Answer to the Comment 6.

5. Comment:

Additionally, several extraordinary claims are made without any citations, and appear to be meant to be taken as self-evidently true. Far from being self-evident, these claims are contradicted by the very work that is described here (see below for specific examples).

Answer:

The extraordinary claims are deleted.

Abstract: "Insertion of functional groups into chemical bonds is an economical and ecofriendly transformation of organic molecules because it leads to the simultaneous formation of two new chemical bonds without significant generation of byproducts." **was replaced by** "Insertion of atoms into chemical bonds is an attractive transformation of organic molecules because it leads to the simultaneous formation of two new chemical bonds. "

Page 2: "Insertion of functional groups into chemical bonds is an economical and ecofriendly transformation of organic molecules because it leads to the simultaneous

formation of two new chemical bonds without significant generation of byproducts.” **was replaced by** “Insertion of atoms into chemical bonds is an attractive transformation of organic molecules because it leads to the simultaneous formation of two new chemical bonds.”.

6. Comment:

Despite the novelty of the transformation, the lack of mechanistic study in this body of work along with the existence alternative methods for preparing the same substrates limit its usefulness to the synthetic community. Regardless of where this work is eventually submitted, major revisions to the manuscript are needed for it to be acceptable as a scientific document.

Answer:

The mechanistic study and a plausible pathway were added:

Page 6: **Mechanistic Studies.** To gain more insight into the transformation, the corresponding dearomatized intermediate **48** and the azidination intermediate **49** were isolated. Both of them can be converted into product **2** under the standard conditions (Fig. 6a). When unsubstituted benzenamine was employed as the substrate, 2.1 equivalents of PhIO were required to facilitate the oxidative dearomatization to generate a quinone imine ketal **51** as the intermediate. However, the reaction of **51** under the standard conditions gave rise to *N*-(4-methoxyphenyl)-4-methylbenzenesulfonamide **52** instead of the insertion product (Fig. 6b). When two azidination intermediates **53** and **54** were mixed and treated with Cu(OTf)₂, the reaction gave rise to compounds **55** and **56**, and the formation of **57** and **58** was not observed (Fig. 6c). This result indicated the migration of the para-alkyl group proceeds via an intramolecular manner.

Fig. 6. Mechanistic Study

A plausible pathway for this transformation was depicted in Fig. 7. PhIO mediates the oxidative dearomatization of para-substituted anilines in methanol to generate cyclohexadienimines. The nucleophilic addition of chiral sulfur ylides to cyclohexadienimines and subsequent cyclization lead to the generation of the spiro intermediates. Rearomatizing to release the tension of the spiro structure is a great driving force for the arrangement. With the aid of a Brønsted or Lewis acid, the migration of the alkyl or the aryl group forms intermediate **I** (path a), while the migration of the methoxy group forms intermediate **II** (path b). Because the positive charge in intermediate **I** can be stabilized by the oxygen atom (intermediate **III**), rearrangement via path a is preferred. Final aromatization delivers the one-carbon insertion products. What's more, when the substituent at the para-position of amino group was the methoxy group, the catalyst would prefer to combine with the methoxy group to make it leave and produce carbocation. The C-C fracture of three-member-ring could make it aromatization, and the resulting intermediate **IV** containing imine cation hydrolyzed rapidly to produce byproduct with no insertion part.

Fig. 7. Plausible reaction pathway

7. Comment:

Abstract: Line 14-16, “Insertion of functional groups into chemical bonds” is vague in meaning and “ecofriendly transformation of organic molecules without significant generation of byproducts” does not appear to be necessarily true, even though it is claimed to be without citation. It should be noted that the method described employs 1 eq of PhIO and 1.2 eq of a chiral sulfonium salt which fatally undercuts the claim that this method is especially and self-evidently environmentally benign.

Answer:

The extraordinary claims are deleted:

Abstract: Line 14-16, “Insertion of functional groups into chemical bonds is an economical and ecofriendly transformation of organic molecules because it leads to the simultaneous formation of two new chemical bonds without significant generation of byproducts.” **was replaced by** “Insertion of atoms into chemical bonds is an attractive transformation of organic molecules because it leads to the simultaneous formation of two new chemical bonds.”.

8. Comment:

Line 25: Omit the word method.

Answer:

Line 25: the word “method” was omitted.

9. Comment:

Lines 33-35 do not cite any source for the veracity of this claim. The sentence itself implies the claim is self-evident, but it is not even true for this very method let alone a general principle.

Answer:

The extraordinary claims are deleted:

Lines 33-35: “Insertion of functional groups into chemical bonds is an economical and ecofriendly transformation of organic molecules because it leads to the simultaneous formation of two new chemical bonds without significant generation of byproducts.” **was replaced by** “Insertion of atoms into chemical bonds is an attractive transformation of organic molecules because it leads to the simultaneous formation of two new chemical bonds.”.

10. Comment:

Line 36: Sometimes the authors say “insertion of functional groups” or just “groups” and sometimes it is “atoms.” This should be changed for the sake of consistency.

Answer:

Line 14: “Insertion of functional groups” **was replaced by** “Insertion of atoms”.

Line 33: “Insertion of functional groups” **was replaced by** “Insertion of atoms”.

Line 36: “Insertion of groups” **was replaced by** “Insertion of atoms”.

11. Comment:

Lines 42-45: Once again, the claim is asserted as a brute-force and/or self-evident fact. The authors appear to be claiming that their method is the “most direct” way of making “nitrogen-containing aromatic molecules” despite the fact that both their starting materials and products are “nitrogen-containing aromatic molecules.” The claim does not even meet the standard of being consistent with the rest of the paper even if it did have external references to validate it, which it does not.

Answer:

The extraordinary claims are deleted:

Abstract: Line 17, “In this context, insertion of atoms into aromatic carbon-nitrogen bonds is one of the most direct methods for the synthesis of nitrogen-containing aromatic molecules” **was replaced by** “In this context, insertion of atoms into aromatic

carbon-nitrogen bonds is an appealing method for the synthesis of nitrogen-containing molecules”.

Line 42-45: “In this context, insertion of atoms into aromatic carbon-nitrogen bonds is one of the most direct methods for the synthesis of nitrogen-containing aromatic molecules” **was replaced by** “In this context, insertion of atoms into aromatic carbon-nitrogen bonds is an appealing method for the synthesis of nitrogen-containing molecules”.

12. Comment:

Line 45 needs the word “the” prior to the word “aromatic”

Answer:

Page 2: Line 45, “Enantioselective one-carbon insertion into aromatic carbon-nitrogen bond” **was removed.** .

13. Comment:

Lines 52-53: the phrase “after further transformations” confuses the intended meaning of the sentence.

Answer:

Lines 52-53: “the nitrogen atom is usually discarded in byproducts after further transformations” **was replaced by** “the nitrogen atom is usually discarded in byproducts”.

14. Comment:

Figure 1: It may be informative to readers to include methods for making chiral benzylic amines so those who may wish to use this method can easily grasp the advantages. Regardless, alternative methods to chiral benzylic amines must be listed in the text or it will be assumed by the readers that the authors are intentionally discouraging a direct comparison to known methods.

Answer:

The introduction of the preparation of chiral benzylic amines and the corresponding references were added. Please see the Answer to the Comment 2.

15. Comment:

Line 66: This sentence means nearly nothing and could be safely removed.

Answer:

Line 66: “The details of this transformation, shown in Figure 2, are quite simple.” **was replaced by** “The underlying principle is shown in Figure 2.”

16. Comment:

Line 67: The language here is imprecise and should be changed. The carbon-nitrogen bond is not activated by dearomatization. A new carbon-nitrogen (π) bond is created by the dearomatization, which can engage the sulfur ylide. This is a misleading shorthand disguised as a formality.

Answer:

Line 67: “Oxidative dearomatization can activate the aromatic carbon-nitrogen bond in anilines by destroying the aromatic system” **was replaced by** “Oxidative dearomatization can transform the aromatic carbon-nitrogen bond in anilines to a carbon-nitrogen double bond by destroying the aromatic system”.

17. Comment:

Line 70: This is the first time it is called a “formal” insertion, but likely it would be more honest to call it a “formal” insertion from the beginning. The fact that it is a formal insertion which relies on oxidative dearomatization to proceed makes Figure 1a seem even more out of place than it already did.

Answer:

Title: “Enantioselective One-Carbon Insertion into Aromatic C-N Bonds via Dearomatization: Conversion of Anilines to Chiral Benzylic Amines” **was replaced by** “Formal Enantioselective One-Carbon Insertion into Aromatic C-N Bonds via Dearomatization: Conversion of Anilines to Chiral Benzylic Amines”.

Abstract: Line 24, “we report an enantioselective one-carbon insertion” **was replaced by** “we report a formal enantioselective one-carbon insertion”.

Line 63: “we report an enantioselective aromatic carbon-nitrogen bond one-carbon insertion reaction” **was replaced by** “we report a formal enantioselective aromatic carbon-nitrogen bond one-carbon insertion reaction”.

18. Comment:

Line 71: Should read “This builds...”

Answer:

Line 71: “This will build” **was replaced by** “This builds”.

19. Comment:

Line 81: “In turn” adds confusion.

Answer:

Line 81: “This might be hampered by the lower reactivity of the ketimino group which in turn is caused by intrinsic electronic and steric factors.” **was removed.**

20. Comment:

Line 82: An effect of reactivity that is caused by “electronic and steric factors” is so broad as to include every possible factor. The sentence therefore reads in an identical way as the following: “The ketimino group might be less reactive because of something.” This need to be corrected.

Answer:

Line 82: “Central to this strategy is the sulfur ylide-mediated aziridination with the ketimino group of the dearomatized intermediate of anilines. This might be hampered by the lower reactivity of the ketimino group which in turn is caused by intrinsic electronic and steric factors.” **was removed.**

21. Comment:

Line 82: “Our initial of the viability” does not make grammatical sense, maybe from a word mistakenly being omitted.

Answer:

Line 82: “Our initial of the viability of the strategy was performed with an achiral sulfonium salt (Fig. 3).” **was replaced by** “To implement this strategy, the rapid oxidative dearomatization of *p*-toluidine **1** was tested by examining various oxidants including *m*CPBA, AcOOH, *t*-BuOOH, H₂O₂, PhI(OAc)₂, and PhIO. PhIO together with methanol as the solvent proved to be the best oxidant conditions for the dearomatization. After removing the methanol in vacuo, the crude dearomatized product was mixed with a solution of achiral sulfonium salt **S1** in acetonitrile in the presence of 1.2 equivalents of NaH, followed by the treatment with 2 equivalents of CF₃COOH (Fig. 3).”.

22. Comment:

Line 84: The authors, when describing their initial viability studies, erroneously fail to mention the oxidation step of *p*-toluidine saying, “reaction of *p*-toluidine **1** with an

ylide.” This simply is not what was done in the study shown, is inconsistent with the proposed mechanism, and must be corrected.

Answer:

Line 84: “Our initial of the viability of the strategy was performed with an achiral sulfonium salt (Fig. 3). Reaction of *p*-toluidine **1** with an ylide derived from **S1** proceeded smoothly and delivered benzylic amines **2** in good yield (71%).” **was replaced by** “To implement this strategy, the rapid oxidative dearomatization of *p*-toluidine **1** was tested by examining various oxidants including *m*CPBA, AcOOH, *t*-BuOOH, H₂O₂, PhI(OAc)₂, and PhIO. PhIO together with methanol as the solvent proved to be the best oxidant conditions for the dearomatization. After removing the methanol in vacuo, the crude dearomatized product was mixed with a solution of achiral sulfonium salt **S1** in acetonitrile in the presence of 1.2 equivalents of NaH, followed by the treatment with 2 equivalents of CF₃COOH (Fig. 3). To our delight, the aziridination and subsequent rearrangement proceeded smoothly and delivered benzylic amines **2** in good yield (71%).”.

23. Comment:

Line 95: The authors list variables tested and, once again, fail to mention anything about the oxidant. It’s necessity in the reaction, for example, must be included. This same logic also applies as completely to the acid-catalyzed rearrangement, which is also erroneously missing from all discussion.

Answer:

The discussion on the details of the oxidation conditions and the acid-catalyzed rearrangement conditions have been added. Please see the Answer to the Comment 1.

24. Comment:

Line 106: The word “they” before “migrate” is not necessary and should be omitted.

Answer:

Line 106: “and they migrate to” **was replaced by** “and migrate to”.

25. Comment:

Line 119: “but” should be “and” for the sentence to make the most sense.

Answer:

Line 119: “but the steric effect” **was replaced by** “and the steric effect”.

26. Comment:

Line 121: Dearomatization step is finally mentioned, albeit in passing.

Answer:

The discussion on the details of the oxidation dearomatization has been added:

Page 4: Line 84, “To implement this strategy, the rapid oxidative dearomatization of *p*-toluidine **1** was tested by examining various oxidants including *m*CPBA, AcOOH, *t*-BuOOH, H₂O₂, PhI(OAc)₂, and PhIO. PhIO together with methanol as the solvent proved to be the best oxidation conditions for the dearomatization.” **was added.**

27. Comment:

Line 136: Please change to “The reaction of a sulfonium.”

Answer:

Line 136: “Reaction of sulfonium salt” **was replaced by** “The reaction of a sulfonium salt”.

28. Comment:

Line 139: Strange symbol typo in this reviewer’s copy.

Answer:

Line 139: the strange symbol typo should be “α-”.

29. Comment:

Line 142: same problem as line 139. Looks like a problem with the alpha symbol.

Answer:

Line 142: the strange symbol typo should be “α-”

30. Comment:

Line 153: “the synthesis of nitrogen-containing aromatic compounds”- both starting materials and products for all of these examples meet this definition. Following this logic, the most direct way to prepare a nitrogen containing aromatic compound is to simply already have one.

Answer:

Line 153: “with a view to the synthesis of nitrogen-containing aromatic compounds” **was removed.**

31. Comment:

Line 176: Along with a discussion of the oxidants screened, mention of the requirement of a substituted aniline as a starting material is omitted but should not be.

Answer:

Line 169: “the aromatic carbon-nitrogen bond in anilines” **was replaced by** “the aromatic carbon-nitrogen bond in para-substituted anilines”.

32. Comment:

Line 176: There is similarly no discussion whatsoever of the role of copper in the reaction. There is no discussion of the necessity of copper in the reaction. The word copper is not even included anywhere in the text. A discussion of the acids for the promotion of the rearrangement is not mentioned anywhere in the text.

Answer:

The discussion on the details of acid-catalyzed rearrangement conditions have been added:

Page 3: Line 69, “Subsequent rearrangement can redevelop the aromaticity and complete the formal enantioselective one-carbon insertion.” **was replaced by** “Subsequent rearrangement might be promoted by a Brønsted or Lewis acid to redevelop the aromaticity and complete the formal enantioselective one-carbon insertion.”.

Page 4: Line 96, “In this way, the yield of **2** was improved to 79% yield and 96% ee” **was replaced by** “NaH proved to be the best base. A variety of Brønsted or Lewis acids shown different activities to promote the rearrangement. When 20 mol% Cu(OTf)₂ was used instead of 2 equivalents CF₃COOH, the yield of **2** was improved to 79% yield and 96% ee”.

A plausible pathway was added:

Page 7: A plausible pathway for this transformation was depicted in Fig. 7. PhIO mediates the oxidative dearomatization of para-substituted anilines in methanol to generate cyclohexadienimines. The nucleophilic addition of chiral sulfur ylides to cyclohexadienimines and subsequent cyclization lead to the generation of the spiro intermediates. Rearomatizing to release the tension of the spiro structure is a great driving force for the arrangement. With the aid of a Brønsted or Lewis acid, the migration of the alkyl or the aryl group forms intermediate **I** (path a), while the migration of the methoxy group forms intermediate **II** (path b). Because the positive charge in intermediate **I** can be

stabilized by the oxygen atom (intermediate **III**), rearrangement via path a is preferred. Final aromatization delivers the one-carbon insertion products. What's more, when the substituent at the para-position of amino group was the methoxy group, the catalyst would prefer to combine with the methoxy group to make it leave and produce carbocation. The C-C fracture of three member ring could make it aromatization, and the resulting intermediate **IV** containing imine cation was gotten which hydrolyzed rapidly to produce byproduct with no insertion part.

Fig. 7. Plausible reaction pathway

Comment:

Line 176: The removal of solvent after the oxidation is also ignored in the main text. This is a multi-step method, and the authors are not served by ignoring the actual protocol.

Answer:

The discussion on the details of the oxidation conditions and the reaction in stepwise manner have been added:

Page 4: Line 79, “To implement this strategy, the rapid oxidative dearomatization of *p*-toluidine **1** was tested by examining various oxidants including *m*CPBA, AcOOH, *t*-BuOOH, H₂O₂, PhI(OAc)₂, and PhIO. PhIO together with methanol as the solvent proved to be the best oxidation conditions for the dearomatization. After removing the methanol in vacuo, the crude dearomatized product was mixed with a solution of achiral sulfonium salt **S1** in acetonitrile in the presence of 1.2 equivalents of NaH, followed by the treatment with 2 equivalents of CF₃COOH (Fig. 3).“ was added.

33. Comment:

Formatting issues in the TOC need to be addressed

Answer:

A TOC was added:

34. Comment:

The following errors occur relatively routinely, and the entirety of the SI should be checked again:

1. The counterion in the MS calc. data is not included in the calculated formula in any of the substrates but should be. For instance $[M+\text{Na}]^+$ calcd. for $\text{C}_{27}\text{H}_{31}\text{NO}_3\text{SNa}$ (sodium atom included).

Answer:

The counter ion in the MS calc. data is included in the calculated formula for all products.

35. Comment:

Multiple substrates have too few carbon signals. HSQC and HMBC should let you know if there are two overlapping, and the fact of a single signal for two different carbon peaks should be noted.

Answer:

All ^{13}C NMR spectra were checked carefully to assign the carbon signals. In some substrates, the aromatic ring which came from sulfur ylide only had substituents with weak ability of electron absorption or electron delivery. So two peaks of the aromatic ring may overlap. ^{13}C NMR 126MHz was tested to get higher resolution, but the overlapping peak can't separate.

Answers for Comments of reviewer 2:

36. Comment:

When ortho-substituted anilines were used, 3-substituted products were dominated in comparison with 5-substituted products. Could the authors explain the reason for the unusual selectivity? Why did the migration happen at a bulkier site? In addition, there

was lack of explanation as fluorinated substrate 17 only led to single regiomer.

Answer:

An explanation for the selectivity was added:

Page 5: Line 118, “The electron-withdrawing effect makes the C-3 position more positively charged compared to the C-5 position, but the steric effect makes the C-3 position more hindered.” **was replaced by** “The electron-withdrawing effect makes the C-3 position more positively charged compared to the C-5 position, and the steric effect makes the C-3 position more hindered than the C-5 position. Due to the strong electron-withdrawing property and the small size of the fluorine atom, the C-3 position is the preferred site for the migration of the 4-methyl group. Therefore, the reaction of 2-fluoro-4-methylaniline only produced the C-3 migration product **17**. When the *ortho*-substituent is a bromine atom or a phenylethynyl group, the relatively weaker electron-withdrawing property and the bigger size led to the formation of mixture of the C-3 or C-5 migration products.”.

37. Comment:

In this manuscript, only semistable sulfur ylides have been tested. The stable ylides bearing an ester or ketone group, are very interesting substrates for this transformation as they will provide chiral phenylglycines.

Answer:

The reaction of stable ylides were examined. However, the corresponding aziridination did not occur under the standard conditions. The ylide bearing an ester group couldn't react with the substrates. The ylide bearing a ketone group reacted with C=C double bond to generate cyclopropane ring.

Answers for Comments of reviewer 3:

38. Comment:

The reactions have been carried out at a relatively small scale. It is important that one example is reported to make 1 g of product to demonstrate the robustness of what appears to be a very useful and interesting reaction.

Answer:

The reaction of *p*-toluidine 1 in a 4 mmol scale was examined:

The reaction produced compound **2** in 78% yield (1.186 g) with 96% ee. The result was added in Fig. 4.

39. Comment:

With respect to the proposed reaction pathway, have the authors evidence for the formation of the proposed aziridine intermediate? E.g., can it be isolated by omitting the final step in the sequence?

Answer:

The aziridine intermediate was isolated, and the conversion of this intermediate to the final product was examined:

Page 6: “To gain more insight into the transformation, the corresponding dearomatized intermediate **48** and the azidination intermediate **49** were isolated. Both of them can be converted into product **2** under the standard conditions (Fig. 6a).” **was added.**

40. Comment:

Line 90: the authors mention the choice of counterion affects the yield of the reaction. Since the reaction proceeds via the ylide (with no counterion) I wonder whether the yield is impacted by the solubility of starting salt. Perhaps the authors could comment.

Answer:

The starting ylide salts we had tested could dissolve well in MeCN and dissolve completely in the reaction. The yield of ylide attacking step was almost unaffected by the starting salts. However, the counterion in ylides could affected the acidity of the reaction system after the addition of Lewis acids or proton acids which decreased the yield.

41. Comment:

Fig. 3 The diagram shows a range of ylides derived from literature known chiral sulfides. Literature references should be provided for each of these sulfides (not currently the case) and an additional reference should be supplied for the best sulfide (Synthesis 2018, 3337 by Aggarwal).

Answer:

The corresponding references **were added** as references 108-113,117.

42. Comment:

The way of drawing Aggarwal’s sulfide might be misleading with respect to the dimethyl substitution at the carbon alpha to sulfur. It would make it less obvious that approach to

one face of the ylide is blocked. Consider changing.

Answer:

The structure of Aggarwal's sulfide was corrected according to reference (J. Am. Chem. Soc. 2010, 1828 by Aggarwal).

43. Comment:

Line 105: in relation to substituents tolerated at the para-position of the aniline, are alkene and alkyne groups tolerated?

Answer:

The start step of reaction is dearomatization. The dearomatization of *p*-alkene aniline or *p*-alkyne aniline couldn't get desired products. The 4-aminostyrenes occurred deoxygenation reaction when oxidated by hypervalent iodine (III) (see *Org. Chem. Front.* 2017, 2156 by Fan). But *p*-alkyne aniline couldn't react with hypervalent iodine compounds.

44. Comment:

Fig.5 The caption refers only to 'benzyl' sulfonium salts but actually the scope goes beyond benzyl (which is excellent). I think the caption should be re-worded to capture this.

Answer:

Page 21, Fig. 5, caption: "Scope of chiral benzyl sulfonium salts." was replaced by "Scope of chiral sulfonium salts."

45. Comment:

Line 146,147: it would be worth noting whether the absolute configuration obtained is in line with expectations based on the literature model for enantioselectivity in S-Ylide aziridinations. Presumably all other configurations are assigned by analogy – a statement to this effect could be included in the supporting information.

Answer:

The absolute configuration obtained is in line with expectations of references (J. Am. Chem. Soc. 2013, 11951 by Aggarwal).

46. Comment:

There are no references in the Supporting Information. There should be some statements about the sources of starting materials and where these are not commercially available a

literature reference should be provided for their synthesis e.g., the various sulfonium salts used. Were all of the anilines commercially available? Are all of the products novel? If not, then references should be provided and a statement that literature data matched should be included.

Answer:

The corresponding references were added as references 1-5 in support information.

Reviewers' Comments:

Reviewer #1:

Remarks to the Author:

The authors have satisfactorily addressed the major issues with their first submission for the Formal Enantioselective One-Carbon Insertion into Aromatic C-N Bonds. Extraordinary claims presented without citations were either removed or appropriately cited, the relevant known preparations of benzylic amines were included, and mechanistic investigations were performed and included. All of these listed changes significantly improve the manuscript and highlight the importance of this contribution in an effective way. The mechanistic studies in particular serve the valuable purpose of illuminating each part of the three-step protocol- a sequence whose details were previously obfuscated. The transformation presented in this manuscript: the formal insertion of a single carbon atom into the C-N bond of para-alkyl substituted aniline to make chiral benzylic amines is more interesting than practical given that it requires both a multistep protocol and stoichiometric chiral material. However, given the novel strategy and unique rearrangement, this manuscript is therefore recommended for publication with the below suggested improvements to the text.

The following suggested changes to the language of the manuscript would further enhance reader's ability to understand and appreciate this work:

Line 68: Sentence is difficult to read. Remove the word "a" before "migration on line 69. On Line 70: change "and" to "and concomitant"

Line 90: change to "protecting group on the aniline nitrogen atom has...." for clarity

Line 177: arrangement should be rearrangement

Line 181: "What's more" is very informal

182: change "the" before methoxy to "a"

Lines 182-186: These two sentences are very confusing. It could be simply stated that in the absence of a para-alkyl substituent, dearomatization occurred to give an acetal, but aziridination and C-C bond cleavage were not observed under standard conditions.

Line 191: "For example" is followed by a comment about scalability but is preceded by a discussion of product derivatization. Consider changing the order of these sentences.

Line 208 "ylides-mediated" should be "ylide-mediated"

Reviewer #2:

Remarks to the Author:

The revised manuscript has been significantly improved, and most of the referees' concerns have been addressed. The isolation of key dearomatized intermediate 48 and the azidination intermediate 49 further increases the integrity of this manuscript. Hence, the referee would recommend the acceptance of this manuscript for publication in Nat Commun as it is.

Here are the details on the point-by-point response to the referees' comments:

Answers for Comments of reviewer 1:

1. Comment:

The following suggested changes to the language of the manuscript would further enhance reader's ability to understand and appreciate this work:

Line 68: Sentence is difficult to read. Remove the word "a" before "migration on line 69.

On Line 70: change "and" to "and concomitant"

Answer:

Page 3, Result section, paragraph 1, line 7: "This builds the nitrogen-substituted benzylic stereocenter and is accompanied by a migration of the para substituent to the meta position and para substitution by a nucleophilic reagent. " **was replaced by** "This builds the nitrogen-substituted benzylic stereocenter and is accompanied by migration of the para substituent to the meta position and concomitant para substitution by a nucleophilic reagent. ".

2. Comment:

Line 90: change to "protecting group on the aniline nitrogen atom has..." for clarity

Answer:

Page 4, Optimization of reaction conditions section, paragraph 1, line 8: "the nature of protecting group on the nitrogen atom the aniline has..." **was replaced by** "the nature of protecting group on the aniline nitrogen atom has...".

3. Comment:

Line 177: arrangement should be rearrangement

Answer:

Page 7, Mechanistic Studies section, paragraph 2, line 6: "arrangement" **was replaced by** "rearrangement".

4. Comment:

Line 181: "What's more" is very informal

Answer:

Page 7, Mechanistic Studies section, paragraph 2, line 10: "What's more" **was removed.**

5. Comment:

182: change “the” before methoxy to “a”

Answer:

Page 7, Mechanistic Studies section, paragraph 2, line 11: “the methoxy group” **was replaced by** “a methoxy group”.

6. Comment:

Lines 182-186: These two sentences are very confusing. It could be simply stated that in the absence of a para-alkyl substituent, dearomatization occurred to give an acetal, but aziridination and C-C bond cleavage were not observed under standard conditions.

Answer:

Page 7, Mechanistic Studies section, paragraph 2, line 10: “What’ s more, when the substituent at the para-position of amino group was the methoxy group, the catalyst would prefer to combine with the methoxy group to make it leave and produce carbocation. The C-C fracture of three member ring could make it aromatization, and the resulting intermediate IV containing imine cation hydrolyzed rapidly to produce byproduct with no insertion part.” **was replaced by** “When the substituent at the para-position of amino group was a methoxy group, dearomatization occurred to give an acetal intermediate, but aziridination and C-C bond cleavage were not observed under standard conditions. ”.

7. Comment:

Line 191: “For example” is followed by a comment about scalability but is preceded by a discussion of product derivatization. Consider changing the order of these sentences.

Answer:

Page 8, Synthetic Applications section, paragraph 1, line 1: “Transformations of the one-carbon insertion products have been explored (Figure 8). For example, the reaction magnifying 50 times can occur normally which can get product in 77% yield and 94% ee. Under...” **was replaced by** “The reaction magnifying 50 times can occur normally which can get product in 77% yield and 94% ee. Transformations of the one-carbon insertion products have been explored (Fig. 8a). For example, under...”.

8. Comment:

Line 208 “ylides-mediated” should be “ylide-mediated”

Answer:

Page 8, Discussion section, paragraph 1, line 6: “sulfur ylides-mediated” **was replaced by** “sulfur ylide-mediated”.